# *Ficus carica* (Linn.) Leaf and Bud Extracts and Their Combination Attenuates Type-1 Diabetes and Its Complications via the Inhibition of Oxidative Stress

**DOI:** 10.3390/foods12040759

**Published:** 2023-02-09

**Authors:** Asmae El Ghouizi, Driss Ousaaid, Hassan Laaroussi, Meryem Bakour, Abderrazak Aboulghazi, Rose Strutch Soutien, Christophe Hano, Badiaa Lyoussi

**Affiliations:** 1Laboratory of Natural Substances, Pharmacology, Environment, Modeling, Health, and Quality of Life (SNAMOPEQ), Faculty of Sciences Dhar El Mahraz, University Sidi Mohamed Ben Abdellah, Fez 30000, Morocco; 2The Higher Institute of Nursing Professions and Health Techniques (ISPITS), Fez 30000, Morocco; 3Department of Chemical Biology, Eure & Loir Campus, University of Orleans, 28000 Chartres, France

**Keywords:** diabetes, *Ficus carica*, antidiabetic effect, antioxidant ability, alloxan

## Abstract

The current work was designed to evaluate the antioxidant activity and antidiabetic effect of *Ficus carica* L. extracts. For that, the leaves and buds of *Ficus carica* L. were analyzed to determine their polyphenolic and flavonoid contents and antioxidant activity. Diabetes was induced by a single dose of alloxan monohydrate (65 mg/kg body weight), then diabetic rats were treated with a dose of 200 mg/kg body weight of the methanolic extracts of *Ficus carica* leaves or buds or their combination for 30 days. Throughout the experiment, blood sugar and body weight were measured every 5 and 7 days respectively. At the end of the experiment, serum and urine were collected for analysis of alanine aminotransferase, aspartate aminotransferase, total cholesterol, triglycerides, creatinine, uric acid, urea, proteins, sodium, potassium, and chloride. Pancreas, liver, and kidney were removed to estimate catalase, glutathione peroxidase, and glutathione activities; lipid peroxidation products were also determined. The results obtained revealed that alloxan has induced hyperglycemia, increased liver and renal biomarkers levels, reduced antioxidative enzymes, and induced lipid peroxidation. However, the treatment with *Ficus carica* leaf and bud extracts, especially their combination, has attenuated all pharmacological perturbations induced by alloxan.

## 1. Introduction

Diabetes mellitus (DM) is a serious public health problem and one of the leading causes of premature morbidity and mortality in the world. The International Diabetes Federation (IDF) in its 9th edition confirms that diabetes is one of the fastest-growing global health emergencies of the 21st century [1]. The report estimates that 463 million people lived with diabetes in 2019 [1]. This number is expected to reach 578 million in 2030 and 700 million in 2045; some countries consider this prevalence as an alarming epidemic state and urgently need international intervention for early diagnosis and efficacious management [1,2]. Diabetes is a glucose homeostasis disorder occurring when the pancreas does not make enough insulin leading to high blood glucose levels, or when the body is unable to use it properly. Moreover, it can be favored by disorders in the use of glucose in tissues, and hereditary, genetic, nutritional habits, sedentary lifestyle, and environmental factors, as well as others, are involved [3]. It is a fairly complex disease, both in terms of understanding its pathophysiology, risk factors, and the genesis of its complications [4].

Diabetes can attack patients’ ability to function, and its complications can be disruptive to patients‘ quality of life [5,6]. This chronic condition acts at different levels leading to macrovascular affections such as cardiovascular diseases and microvascular affections including nephropathy, retinopathy, neuropathy, and so on [5,7]. Patients who are not treated correctly could develop serious health complications highly associated with the risk of morbidity and mortality.

Nowadays, conventional interventions and treatment strategies based on the single or combined use of chemical hypoglycemic and anti-hyperglycemic drugs, e.g., thiazolidinediones, metformin, sulphonylureas, sodium-glucose co-transporter-2 (SGLT-2) inhibitors, glucagon-like peptide-1 (GLP-1) agonists or insulin, and dipeptidyl peptidase-4 (DPP-IV) enzyme inhibitors are the mainstays of diabetes management [8]. However, chronic treatment using orthodox medicine could be associated with numerous health complications/undesirable side effects including nausea, abdominal pain, hypoglycemia, gastrointestinal disturbances, lactic acidosis, and body weight gain [9]. Moreover, the conventional management/treatment of diabetes and its serious complications requires a heavy and substantial economic cost for both developing and developed countries worldwide [10]. Therefore, research for new safer and bio-effective anti-diabetic agents from natural origins is needed to regulate chronic hyperglycemia and to prevent diabetes mellitus, diabetic nephropathy, hepatotoxicity, and other associated cardio-metabolic disorders.

Herbal remedies are traditionally and widely used all over the world to control hyperglycemia [11]. Therefore, many medicinal plants have been studied to discover potential hypoglycemic molecules, as many existing drugs are derived from plants [12,13]. The World Health Organization (WHO) has recommended traditional herbs as an excellent candidate and valuable alternative for the management of diabetes due to their low toxicity [14]. In that same context, medicinal plants are used primarily by the Moroccan population for the treatment of diabetes [15,16,17].

Fig or *Ficus carica* (L.) (FC) is known in Morocco by its vernacular name Kermous and 194 genotypes are known [18]. It is belonging to the Moraceae family with more than 1400 species classified into approximately 40 genera characterized by milky latex in all parenchymatous tissue [19]. This tree has been cultivated for a very long time in a good number of countries. They are found primarily in tropical and subtropical temperate zones, are widely cultivated, and are considered an edible product in Mediterranean areas [20,21].

Several studies have reported that different parts of the FC tree contain a high concentration of bioactive compounds, notably cyanidin, chlorogenic acid, rutin, luteolin, and catechin, among others [22,23]. In particular, it has been reported that *Ficus carica* leaves contain sugars, pectin, tannins, vitamin C, trace elements, and a large number of flavonoids [24,25]. In addition to its high nutritive value, the different parts of the FC tree are largely used in folk medicine systems such as Siddha Ayurveda and Unani as a mild natural laxative, diuretic, expectorant agent, deobstruent in liver and spleen diseases, and as an anti-inflammatory agent [21,26].

Gemmotherapy, also called phytoembryo-therapy or meristemo-therapy, has been defined as a therapeutic method based on the use of embryonic and meristematic plant parts, such as young shoots, rootlets, and buds, instead of mature parts for the treatment and prevention of acute and chronic diseases [27]. Because of their bioactive composition, bud derivatives are the most important products in this category and represent one of the supply chains investigated all over the world [28]. Although gemmotherapy is therapeutically important, there is a lack of scientific studies on this natural and potent source of bioactive compounds.

The current study aims to evaluate the therapeutic effect of two parts of *Ficus carica* (leaves and buds) against type 1 diabetes and its complications induced by alloxan injection.

## 2. Materials and Methods

### 2.1. Extracts Preparation

*Ficus carica* (L.), (white variety: Lembdar labiad) was identified and collected from Sefrou city in Morocco (33°50′57″ N, 4°51′40″ W) in 2020. The samples are available at the herbarium of the Faculty of Sciences Dhar El Mehraz, Fez, Morocco, under the following voucher names: (WL20) for *Ficus carica* (L.) leaves and (WB21) for *Ficus carica* (L.) buds.

The leaves were cleaned and air-dried for 15 days, then turned into fine powder. Fresh buds were put in the fridge at 4 °C until it was used to prepare extracts used in the experiments.

The extracts of the leaves and buds of *Ficus carica* (L.) were prepared as follows: 500 g of the plant (dried leaves or fresh buds) were macerated in hydro-methanolic solution (70% *v*/*v*) for one week under agitation at 150 rpm at room temperature. The obtained solution was filtered using a Whatman filter paper no. 1 and then concentrated in a rotary evaporator (Büchi R-210, Flawil, Switzerland) at 40 °C under reduced pressure to obtain a solid residue [29]. This later was dissolved in distilled water to obtain the desired concentrations selected for the in vitro and in vivo experiments.

### 2.2. Quantification of Phenolic Content (TPC)

The colorimetric method using the Folin–Ciocalteu reagent was used to determine the total phenolic content in different extracts of the leaves and buds of *Ficus carica* (L.) as previously described by Bakour et al. [30]. A total of 100 µL of plant extract was mixed with 500 μL of Folin–Ciocalteau (0.2 N) reagent and 400 μL of sodium carbonate solution. Gallic acid was used as a standard to achieve the calibration curve, Absorbances of the reaction mixtures were read at 760 nm using the Perkin Elmer Lambda 40 UV/VIS spectrophotometer. The results obtained were expressed as milligrams of gallic acid equivalent per gram of extract (mg GAE/g).

### 2.3. Quantification of Flavonoids Content (TFC)

Total flavonoid content was quantified according to the protocol described by Kong et al. [31]. Briefly, 100 µL of plant extract, 150 µL of sodium nitrite (5%), and 150 µL of aluminum trichloride (10%) were mixed. Then, after 5 min, 200 µL of hydroxide sodium (1%) was added to the mixture. The mixture was vortexed and incubated for one hour. The absorbance was measured at 510 nm using the Perkin Elmer Lambda 40 UV/VIS spectrophotometer. The results obtained were expressed as mg equivalent quercetin per gram of extract (mg QE/g).

### 2.4. Antioxidant Activity

The antioxidant activity of different extracts was determined using the DPPH assay previously described by Laaroussi et al. [32]. Briefly, 25 µL of leaf and bud extracts, respectively, were mixed with 875 µL of DPPH solution (63.4 µM). The mixtures were incubated for one hour in darkness. The absorbance was read at 517 nm using the Perkin Elmer Lambda 40 UV/VIS spectrophotometer. The scavenging ability of different extracts was estimated using the following formula:$$\%\;\mathrm{inhibition}=\left(\frac{A0-A1}{A0}\right)*100$$

A0: Absorbance of the control

A1: Absorbance of the extract

The IC_50_ DPPH was deduced from the graph of the percentage of inhibition of the plant extract.

### 2.5. Ethical Approval

The experiment was conducted in the Laboratory of Natural substances, Modeling, Optimization Pharmacology, Environment, Modeling, Health and Quality of Life (SNAMOPEQ), Faculty of Sciences Dhar El Mahraz, University Sidi Mohamed Ben Abdellah, Fez, Morocco, under ethical number L.20.USMBA-SNAMOPEQ 2020-03. All applicable international, national, and/or institutional guidelines for the care and use of animals were followed.

### 2.6. Experimental Design and Treatment Schedule

This study was conducted on normal male Wistar rats (*n* = 48) weighing 220–230 g each at 8 weeks of age. The animals were obtained from the Animal House-Breeding Center of the Biology Department in the Faculty of Science and Technology of Fez, Morocco. The rats were acclimatized for one week in standard environmental conditions (23 ± 3 °C with 12 h light/dark cycles); and were fed with standard rodent chow and water ad libitum. Four groups served as control and four groups were used to induce diabetes type 1 through a single intravenous injection of alloxan monohydrate (65 mg/kg BW) prepared in normal saline. After 72 h passed, only rats with serum blood glucose ≥ 250 mg/dL were considered diabetics and used in the experiments.

The experimental design was carried out as follows:

Group 1 (NC): six healthy rats (non-diabetic) received distilled water (10 mL/kg BW).

Group 2 (NC + L): six healthy rats (non-diabetic) received orally a dose of (200 mg/kg BW) of the hydro-methanolic extract of *Ficus carica* (L.) leaves dissolved in distilled water.

Group 3 (NC + B): six healthy rats (non-diabetic) received orally a dose of (200 mg/kg BW) of the hydro-methanolic extract of *Ficus carica* (L.) buds dissolved in distilled water.

Group 4(NC + LB): six healthy rats (non-diabetic) received orally a combination dose of (100 mg/kg BW) of the hydro-methanolic extract of *Ficus carica* (L.) leaves and 100 mg/kg BW) of the hydro-methanolic extract of *Ficus carica* (L.) buds.

Group 5 (DC): six diabetic rats received distilled water (10 ml/kg BW).

Group 6 (DC + L): six diabetic rats were treated orally with a dose of (200 mg/kg BW) of the hydro-methanolic extract of *Ficus carica* (L.) leaves dissolved in distilled water.

Group 7 (DC + B): six diabetic rats were treated orally with a dose of (200 mg/kg BW) of the hydro-methanolic extract of *Ficus carica* (L.) buds dissolved in distilled water.

Group 8 (DC + LB): six diabetic rats were treated orally with a combination dose of (100 mg/kg BW) of the hydro-methanolic extract of *Ficus carica* (L.) leaves and 100 mg/kg BW) of the hydro-methanolic extract of *Ficus carica* (L.) buds. All treatments lasted for 30 days.

### 2.7. Biochemical Analysis

Throughout the experiment, blood sugar was measured every 5 days for 30 days, using the dextro-test method, and towards the end, serum samples were analyzed for liver enzymes including aspartate aminotransferases (AST) using kit number 7D81-20 and alanine aminotransferases (ALT) using kit number 7D56-20. During lipid profiling, we determined total cholesterol (TC) using kit number 7D73-20 and the cholesterol oxidase/POD method, and triglycerides (TG) using kit number 7D74-20 and the lipase/GK/POD method.

To explore the kidney function, serum kidney parameters (urea, creatinine) were analyzed, 24 h urine was collected and the creatinine and uric acid levels, total protein, sodium (Na^+^), potassium (K^+^), and chloride (Cl^−^) were analyzed in normal and diabetic rats using the ion-selective potentiometry method (Architect c8000i biochemistry analyzer) with kits under numbers 1E49-01, LN9D28-02, and 1E48-20, respectively.

### 2.8. Blood Sampling and Tissue Preparation

To obtain sufficient volume and quality of blood samples, the refined retro-orbital bleeding (ROB) method using the lateral approach was performed as described by Ashish Sharma et al. [33]. For this purpose, sterile glass Pasteur transfer pipettes (with flat edges) were used. Animals were mildly sedated with diethyl ether (inhalation route), and the eyelid was pulled back to proptose the eye. The flat edge of the pipette was placed at the lateral canthus and was oriented toward the back of the head at an angle of 45° to the sagittal and coronal planes. Then it was twisted gently with pressure against the orbital bone just in front of the zygomatic arch until blood flowed from the capillaries draining the orbital sinus. This way, capillary motion draws blood into the tube. Collected Blood samples were centrifuged at 3000 rpm for 15 min, and serum was isolated and stored at −20 °C until biochemical analysis.

At the end of the experiment, all rats were sacrificed using the decapitation method without chemical anesthetics (collected organs must be fresh and free of chemicals), and the kidney, pancreas, and liver were quickly removed and washed with ice-cold saline solution. Therefore, each organ was finely crushed and homogenized in cold phosphate-buffered (0.1 M; pH 7.4) and centrifuged at 8000× *g* for 20 min at 4 °C. The supernatant was collected and stored at −20 °C to analyze the oxidative stress parameters [34].

### 2.9. Catalase Activity in the Pancreas, Liver, and Kidneys of Normal and Diabetic Rats

Catalase (CAT) activity was determined using the method described by Aebi et al. [35]. A decrease in absorbance due to H_2_O_2_ degradation was monitored spectrophotometrically at 240 nm for 1 min, and the activity was expressed as μmol H_2_O_2_/min/mg protein.

### 2.10. Glutathione (GSH) Activity in the Pancreas, Liver, and Kidneys of Normal and Diabetic Rats

GSH levels were measured as described by Ellman et al. [36]. Briefly, 3 mL of sulfosalicylic acid (4%) was added to 500 mL of the homogenate liver, pancreas, and kidney tissues. The mixture was centrifuged at 2500× *g* for 15 min and then the prepared Ellman’s reagent was added to 500 mL of the supernatant. The absorbance was measured at 412 nm after 10 min. The total GSH content was expressed as μg/g of tissue.

### 2.11. Peroxidase Activity in the Pancreas, Liver, and Kidneys of Normal and Diabetic Rats

Glutathione peroxidase (GPx) activity was estimated according to the method of Flohé et al. [37]. GPx activity was expressed as moles of GSH oxidized/min/mg protein.

### 2.12. Malondialdehyde MDA Levels in the Pancreas, Liver, and Kidneys of Normal and Diabetic Rats

The formation of lipid peroxidation products (malondialdehyde) was quantified in the liver, pancreas, and kidney tissues using the thiobarbituric acid-reactive substances (TBARSs) method, as reported previously by Kassan et al. [38], and the absorbance was measured at 532 nm. Results were expressed as malondialdehyde (MDA) concentration (nmol/g tissue).

### 2.13. Statistical Analysis

Statistical comparisons between the groups were performed with a one-way analysis of variance (ANOVA) followed by the Tukey test, and the comparison between rat groups was made by *t*-test using GraphPad Prism ® software (version 5.0; GraphPad Software, Inc., San Diego, CA, USA). Data were represented as mean ± SD (* *p* < 0.05, ** *p* < 0.01, and *** *p* < 0.001).

## 3. Results

### 3.1. Quantification of Phenolic and Flavonoid Contents and Antioxidant Activity

The results obtained from the quantification of the phenolic and flavonoid content and the antioxidant activity of different extracts of the leaves and buds of FC are shown in Table 1. Values of total polyphenolic content are higher in FC buds than in leaves (148.17 ± 8.54 in buds vs. 74.58 ± 3.02 in leaves), whereas the contrary was observed for flavonoid content (1.01 ± 0.02 in buds vs. 11.47 ± 0.01 in leaves).

As shown in Table 1, the IC_50_ DPPH value of *Ficus carica* buds is lower than the IC_50_ DPPH value obtained by *Ficus carica* leaves, (0.26 ± 0.05 vs. 0.31 ± 0.04 respectively).

### 3.2. Biological Assessments

#### 3.2.1. Impact of Different Treatments on Body Weight

Table 2 displays the change in body weight of rats during the experimental period. The body weight of the non-diabetic groups developed normally, whereas in the diabetic group, the body weight decreased significantly during the experimental period. This decrease was protected in the treated diabetic groups, especially those treated with bud extract and the combination of bud and leaf extract.

#### 3.2.2. Impact of Different Treatments on Blood Sugar Levels

The results of the variation in blood sugar levels during the experiment period are shown in Table 3. The change in blood sugar level was significantly controlled by administering different extracts of leaves and buds and restoring glycemia to the normal range (≈1 g/L) after 30 days of treatment, while the positive control showed an increase in blood glucose level with values ranging from 2.92 ± 0.50 g/L on day 0 to 4.19 ± 0.56 g/L on day 30.

#### 3.2.3. Impact of Different Treatments on Lipid Profile

The results of the effect of alloxan and the oral administration of 200 mg/kg of FC leaf and bud extracts and their combination for 4 weeks on the total cholesterol level of the different groups compared to the control group in Wistar rats are summarized in Figure 1 and showed that alloxan administration significantly increased the total cholesterol levels in rats’ blood. These results showed also a significant lowering effect of leaf and bud extract administration on this parameter. Results showed also a significant increase in TG levels in the diabetic control group compared to the normal control group, whereas the treatment with 200 mg/kg of fig leaf and bud extracts has alleviated this effect more importantly than leaf extract only.

#### 3.2.4. Impact of Different Extracts on Liver Enzymes

The effect of alloxan and oral administration of 200 mg/kg of *Ficus carica* L. leaf and bud extracts and their combination for 4 weeks on ALT and AST levels in diabetic and normal groups is presented in Figure 2 and showed that the liver enzymes ALT and AST are significantly increased in the diabetic non-treated group compared to the normal control group, whereas the treatment with *Ficus carica* L. leaf and bud extracts and their combination has protected the liver from the damage induced by hyperglycemia.

#### 3.2.5. Impact of Different Treatments on Serum and Urinary Kidney Parameters

The results presented in Figure 3 summarize the effect of alloxan and oral administration of 200 mg/kg of *Ficus carica* L. leaf and bud extracts and their combination for 4 weeks on serum creatinine and urea levels and show a significant increase in the creatinine and urea level in the diabetic control group compared with the normal control group. The treatment with *Ficus carica* leaf and bud extracts and their combination has prevented the renal damage caused by diabetes toxicity. Results showed that the most pronounced effect is attributed to the combination of leaf and bud extracts.

Figure 3 displays the results of the effect of alloxan, and oral administration of 200 mg/kg of *Ficus carica* L. leaf and bud extracts and their combination for 4 weeks on urinary kidney parameters. The results indicate a significant decrease in urinary creatinine and an important elimination of urinary sodium, potassium, and chlorides besides an important protein and uric acid leak in the diabetic control rats. In contrast, the quotidian treatment by *Ficus carica* L. leaf and bud extracts and their combination has significantly corrected these pathologic changes.

#### 3.2.6. Impact of Different Treatments on the Kidney, Pancreas, and Liver Enzymatic Antioxidants and Lipid Peroxidation

The status of oxidative stress in the kidney, liver, and pancreas of diabetic and non-diabetic rats was investigated by measuring protein, MDA, catalase, GSH, and GPx levels. The results presented in Table 4, Table 5 and Table 6 showed that the catalase, GPx, and GSH levels were significantly decreased in the three organs of the diabetic groups compared with the normal groups, however, the daily oral administration of *Ficus carica* L. leaf and bud extracts and their combination has improved those biomarkers. An important increase in proteins and MDA levels in diabetic groups for the three organs has been also noticed which decreased after treatment with the different extracts.

## 4. Discussion

*Ficus carica* is a well-known plant for its unsuspected salutary effects and its multiuse in folk medicine [39]. It possesses a wide spectrum of biological properties including antipyretic, anti-inflammatory, antispasmodic, and anti-constipation effects [40]. In the present study, two parts of FC (leaves and buds) were tested against diabetes and its complications using methanolic extracts. As mentioned in the results part, the plant has a beneficial effect on all physiological parameters monitored in Wistar rats rendered diabetic by a single injection of alloxan.

As a medicinal plant, FC contains bioactive compounds that have therapeutic or preventive effects against several toxic agents [39,41,42]. Phytochemical screening of FC revealed the presence of a cocktail of biochemical molecules including phenolic compounds, flavonoids, anthocyanins, organic acids, and phytosterols [39]. In the current work, the results showed that the buds and leaves were rich in phenolic compounds. These results are higher than those evoked by Toma et al. [26] and are in line with the values obtained in the study of Iftikhar et al. [43], whereas the total flavonoid content in buds and leaves was lower than those reported by Toma et al. [26] and Iftikhar et al. [43]. With the substantial antioxidant ability observed in the current study, both leaf and bud extracts demonstrated appreciable free radical scavenging activity.

Alloxan is a diabetogenic chemical agent that degenerates and regresses ß pancreatic-cells size [43]. The intravenous injection of Wistar rats with alloxan disrupts glucose homeostasis, which depends on numerous factors including glycogen metabolism, insulin sensibility, glycolysis, and gluconeogenesis [44]. The administration of FC extracts ameliorates the upsurge of glycemia induced by alloxan. In particular, the combination of leaf and bud extracts showed a remarkable synergistic effect to decrease blood sugar levels. It has been proved that the aqueous extract of *Ficus carica* leaves possesses a hypoglycemic effect on streptozotocin-induced diabetic rats [45], and it enhances the production and the release of insulin [46]. In addition to that, a preliminary evaluation of the ability of *Ficus carica* to inhibit carbohydrates hydrolyzing enzymes revealed a strong inhibition activity [47,48], which significantly improves metabolic disorders.

On the other hand, alloxan augments cholesterol, triglyceride, and LDL-C levels as previously documented by numerous studies [49,50]. In the present study, treatment of alloxanized hyperglycemic rats with different extracts produced a marked improvement in the lipid profile by reducing total cholesterol and triglyceride levels. Recent studies showed that the administration of fig tree leaves normalized the lipid profile in the animal model [51,52]. Additionally, our results showed a remarkable hepatoprotective effect of *Ficus carica* leaf and bud extracts and their combination in diabetic-treated groups by reducing the transaminases enzymes release. This effect was similar to that observed in previous in vivo hepatoprotective studies showing that *Ficus carica* leaf extract could reverse liver injury induced by different agents [53,54,55]. In the same manner, our results showed a great protective effect against kidney alloxan-induced damage evidenced by a rise in the creatinine and urea blood levels, electrolytes, uric acid, and protein losses in urine. These results align with previous studies revealing the nephroprotective effect of *Ficus carica* extracts against kidney injuries resulting in streptozotocin-induced diabetes in rats [56,57].

In diabetes with persistent hyperglycemia, an overload of ROS is evident which leads to the production of inflammatory factors, lipid peroxidation, protein glycation, and alteration of the cell membrane. Besides this, hyperglycemia aggravates oxidative stress through the alteration of endogenous antioxidant systems notably, the inactivation of antioxidant enzymes such as catalase, GSH, GPx, and SOD [58,59]. Effectively, in this present work, the results showed that alloxan has induced persistent hyperglycemia which induced lipid peroxidation evidenced by the significant increase in MDA levels, and inflammation revealed by the increase in protein levels in the pancreas, liver, and kidney. Furthermore, our study showed that alloxan-induced hyperglycemia has altered the enzymatic antioxidant status by a reduction in the most critical antioxidant enzymes (catalase, GSH, and GPx) synthesis in organs. *Ficus carica* is well known for its protective effects against inflammation and oxidative stress, evidenced by plenty of studies [54,56].

FC bud extracts were previously evaluated against the toxic effect of tebuconazole (a pesticide) on the liver and testes of bats. The results revealed that *Ficus carica* extract attenuates lipid peroxidation, and protein oxidation, and increases antioxidant enzymes in organs [60]. It has been shown that the administration of FC minimizes the genetic perturbations induced by 5-fluorouracil through upregulating cardiac Bcl_2_ mRNA and downregulating troponin gene and reestablishes antioxidants of cardiac and kidney tissues [61]. Multiple phenolic compounds have been detected in FC with different amounts, such as caffeic acid, ferulic acid, syringic acid, chlorogenic acid, quinol, catechin, quercetin, myricetin, and gallic acid [54]. These molecules exhibit high antioxidant potential and present potential beneficial properties through a synergistic effect.

## 5. Conclusions

Overall, our finding proved the antidiabetic effect of long-term treatment with *Ficus carica* bud and leaf extracts. The results of the study showed a significant decrease in blood glucose in diabetic rats. The outcomes of the biochemical parameters supported the antidiabetic effect of *Ficus carica* bud and leaf extracts by decreasing the serum level of liver enzymes; the results also showed an improvement in renal function. According to the antioxidant activity and the considerable amount of phenolic and flavonoid contents, *Ficus carica* bud and leaf extracts have alleviated oxidative stress status by increasing the endogenous antioxidant enzyme activities and decreasing lipid peroxidation levels in pancreatic, renal, and liver tissues. Our finding suggests that *Ficus carica* bud and leaf extracts can help in the treatment of diabetes and in the prevention of its related complications. Further investigations related to the above findings are needed for the discovery of new plant-derived antidiabetic molecules for future drug development.

## Figures and Tables

**Figure 1 foods-12-00759-f001:**
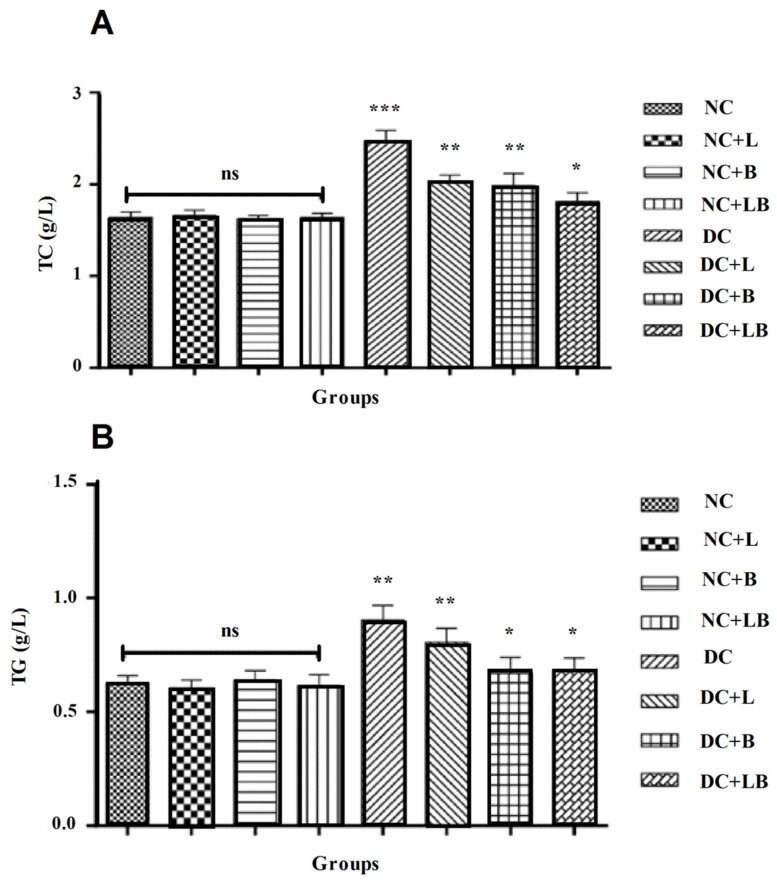
Impact of the daily administration of leaf and bud extracts and their combination on cholesterol levels (**A**) and triglycerides (**B**) of different groups. Data are presented as mean ± SD; statistical analysis was performed between the normal control group (NC) and all remaining groups. (The significance started at * *p* < 0.05; ** *p* < 0.01; *** *p* < 0.001).

**Figure 2 foods-12-00759-f002:**
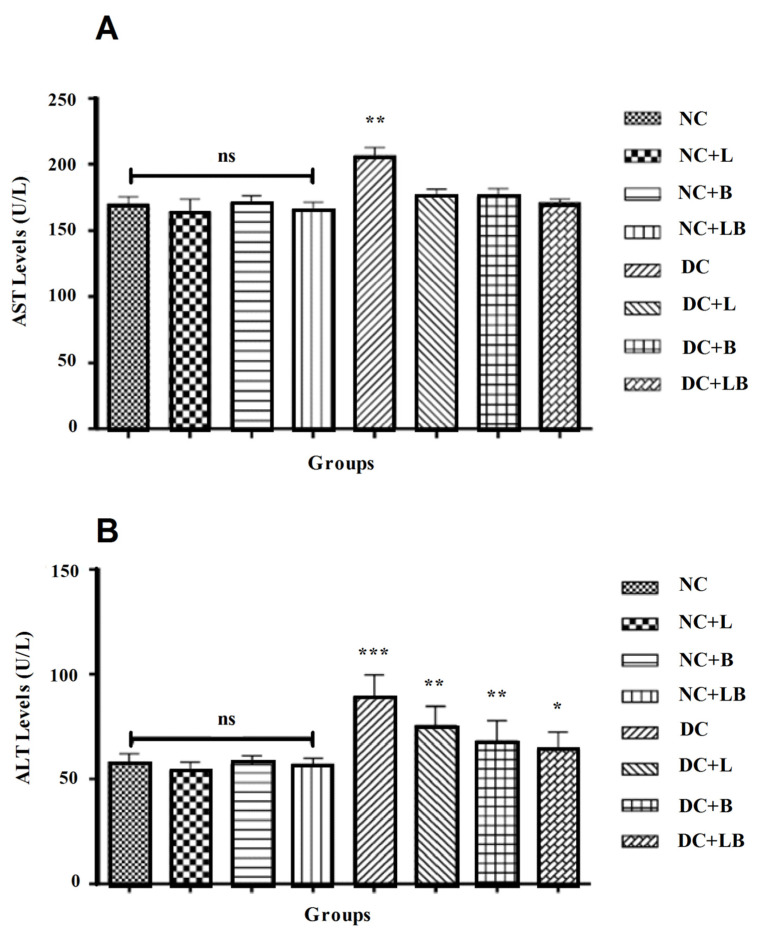
Impact of the daily administration of leaf and bud extracts and their combination on AST (**A**) and ALT (**B**) levels of different groups. Data are presented as mean ± SD; (*) comparison between the normal control group (NC) and all remaining groups. (The significance started at: * *p* < 0.05; ** *p* < 0.01; *** *p* < 0.001).

**Figure 3 foods-12-00759-f003:**
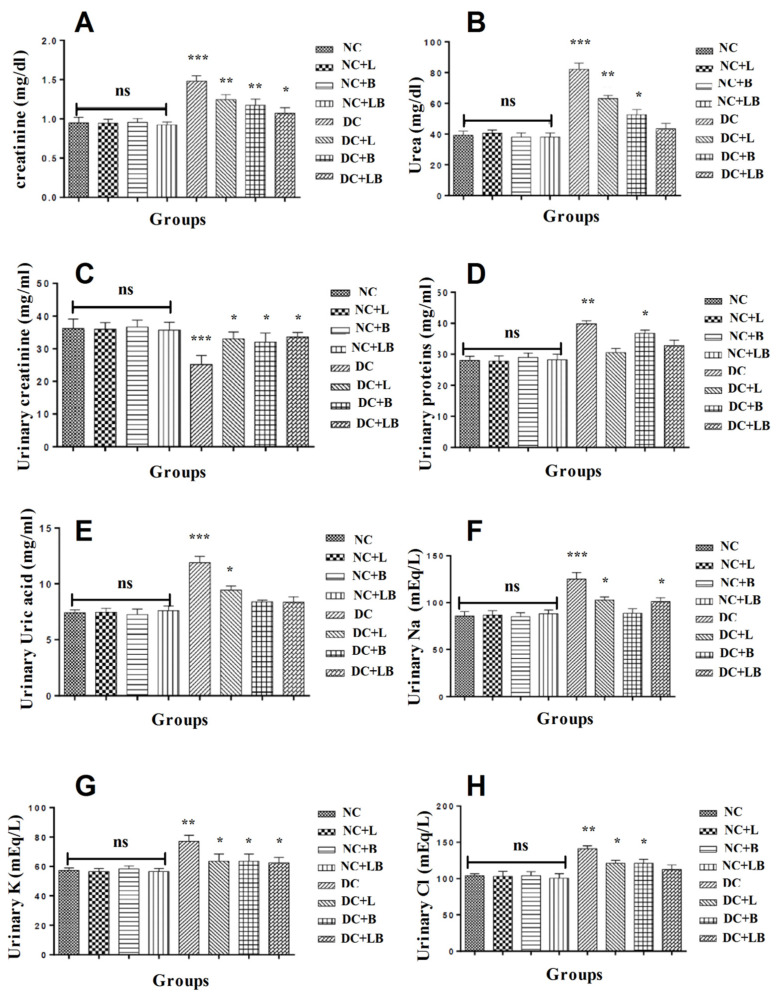
Impact of the daily administration of leaf and bud extracts and their combination on serum and urinary kidney parameters: (**A**) serum creatinine levels; (**B**) serum urea levels; (**C**) urinary creatinine levels; (**D**) urinary protein levels; (**E**) urinary uric acid levels; (**F**) urinary sodium levels; (**G**) urinary potassium levels; (**H**) urinary chloride levels of different groups. Data are presented as mean ± SD; (*) comparison between the normal control group (NC) and all remaining groups. (The significance started at: * *p* < 0.05; ** *p* < 0.01; *** *p* < 0.001).

**Table 1 foods-12-00759-t001:** Phytochemical content and antioxidant activity of different extracts of leaves and buds of *Ficus carica*.

	TPC mg GAE/g DW	TFC mg QE/g DW	DPPH IC_50_ mg/mL
Buds of *Ficus carica*	148.17 ± 8.54 ^a^	1.01 ± 0.02 ^c^	0.26 ± 0.05 ^b^
Leaves of *Ficus carica*	74.58 ± 3.02 ^b^	11.47 ± 0.01 ^a^	0.31 ± 0.04 ^a^

Values in the same column followed by different letters are significantly different.

**Table 2 foods-12-00759-t002:** Body weight changes of normal and diabetic rats during the experimental period (4 weeks).

	Body Weight (g)				
Days of Treatment	Day 0	Day 7	Day 14	Day 21	Day 30
NC	220 ± 5	210 ± 6	224 ± 11	226 ± 9	230 ± 12
NC + L	222 ± 7	224 ± 5	226 ± 9	229 ± 10	232 ± 11
NC + B	224 ± 8	225 ± 3	229 ± 8	230 ± 10	232 ± 9
NC + LB	228 ± 7	230 ± 7	232 ± 10	230 ± 6	233 ± 7
DC	230 ± 6	220 ± 8	200 ± 14 ***	197 ± 8 ***	189 ± 6 ***
DC + L	231 ± 7	229 ± 8	227 ± 5	225 ± 7	220 ± 8 *
DC + B	224 ± 5	225 ± 7	222 ± 8	220 ± 6	218 ± 7
DC + LB	225 ± 7	224 ± 5	220 ± 7	218 ± 9	219 ± 8

The statistical analysis was performed between the days of treatment and day 0. (The significance started at * *p* < 0.05; *** *p* < 0.001).

**Table 3 foods-12-00759-t003:** Blood sugar levels of normal and diabetic rats during the experimental period (4 weeks).

	Blood Glucose (g/L)						
Days of Treatment	Day 0	Day 5	Day 10	Day 15	Day 20	Day 25	Day 30
NC	1 ± 0.5	0.95 ± 0.12	1.02 ± 0.10	0.99 ± 0.14	1.10 ± 0.08	1.12 ± 0.08	0.99 ± 0.09
NC + L	1.10 ± 0.7	0.98 ± 0.11	0.93 ± 0.11	0.97 ± 0.18	1.16 ± 0.06	1.10 ± 0.10	1.05 ± 0.05
NC + B	0.96 ± 0.20	0.99 ± 0.07	1.18 ± 0.05	1.02 ± 0.10	1.11 ± 0.06	1.02 ± 0.07	1.03 ± 0.10
NC + LB	1.12 ± 0.5	1.10 ± 0.10	1.08 ± 0.04	1.09 ± 0.06	1.07 ± 0.05	1.09 ± 0.04	0.98 ± 0.06
DC	2.92 ± 0.50	3.99 ± 0.15 ***	4 ± 0.21 ***	3.98 ± 0.36 ***	4.20 ± 0.41 ***	3.92 ± 0.45 ***	4.19 ± 0.56 ***
DC + L	3.89 ± 0.62	3.52 ± 0.12 *	2.99 ± 0.10 *	2.88 ± 0.62 **	2.72 ± 0.59 ***	1.98 ± 0.36 ***	1.90 ± 0.52 ***
DC + B	3.91 ± 0.38	3.88 ± 0.45	3.62 ± 0.48 ***	2.11 ± 0.25 ***	1.99 ± 0.25 ***	1.62 ± 0.39 ***	1.27 ± 0.30 ***
DC + LB	3.95 ± 0.40	3.20 ± 0.30 *	3.11 ± 0.25 ***	2.40 ± 0.90 ***	1.92 ± 0.60 ***	1.78 ± 0.36 ***	1.62 ± 0.40 ***

The statistical analysis was performed between the days of treatment and day 0. (Significance started at * *p* < 0.05; ** *p* < 0.01; *** *p* < 0.001).

**Table 4 foods-12-00759-t004:** Effect of the different treatments on the oxidative parameters in the kidney (mean ± SD).

Variables in the Kidney	Interventions							
	NC	NC + L	NC + B	NC + LB	DC	DC + L	DC + B	DC + LB
Proteins (mg/g org)	6.15 ± 0.5	5.09 ± 0.23	5.04 ± 0.17	5.85 ± 0.37	8.03 ± 0.53 ^a^***	5.54 ± 0.45 ^a^**^, b^***	4.76 ± 0.19 ^b^***	5.06 ± 0.54 ^b^***
Catalase (µmol H_2_ O_2_/min/mg pr)	32.19 ± 2.05	29.23 ± 1.41	21.23 ± 1.58	30.18 ± 2.21	15.90 ± 2.14 ^a^***	20.45 ± 1,21 ^a^**^,b^***	31.96 ± 2.45 ^b^***	29.68 ± 1.67 ^b^***
GSH (µg/g org)	220.13 ± 5.23	211.09 ± 3.54	229.19 ± 5.09	226.12 ± 4.23	59.78 ± 4.09 ^a^***	190.67 ± 4.06 ^a^**^,b^***	231.22 ± 4.17 ^a^*^,b^***	229.89 ± 5.67 ^b^***
GPx (nmol GSH/min/mg pr)	15.34 ± 1.54	17.19 ± 1.78	18.09 ± 1.06	15.23 ± 0.98	6.67 ± 0.31 ^a^***	13.27 ± 1.45 ^a^**^b^***	15.98 ± 1.69 ^b^***	14.54 ± 0.78 ^a^**^,b^***
MDA (nmol/g org)	24.67 ± 2.09	23.96 ± 2.23	23.45 ± 1.89	24.06 ± 1.22	85.09 ± 3.52 ^a^***	52.16 ± 2.34 ^a^***^,b^***	20.56 ± 4.81 ^a^*^,b^***	22.62 ± 2.61 ^a^*^,b^***

NC: Normal control. NC + L: Normal control treated by leaf extract. NC + B: Normal control treated by bud extract. NC + LB: Normal control treated by the combination of leaf and bud extracts. DC: Diabetic control, Dc+ L: Diabetic + leaf extract. DC + B: Diabetic + buds extract, DC + LB: Diabetic + the combination of leaf and bud extracts. The values are expressed in mean ± SD value, * *p* < 0.05, ** *p* < 0.01, and *** *p* < 0.001; ^a^ comparison between the control group and all other groups; ^b^ comparison between the DC group and the remaining groups.

**Table 5 foods-12-00759-t005:** Effect of the different treatments on the oxidative parameters in the liver (mean ± SD).

Variables in the Liver	Interventions							
	NC	NC + L	NC + B	NC + LB	DC	DC + L	DC + B	DC + LB
Proteins (mg/g org)	8.14 ± 1.63	7.97 ± 1.63	8.53 ± 1.56	7.77 ± 2.05	12.02 ± 2.8 ^a^***	9.56 ± 2.04 ^a^*^, b^***	9.06 ± 1.91 ^b^***	8.63 ± 2.06 ^b^**
Catalase (µmol H_2_ O_2_/min/mg pr)	30.14 ± 0.52	28.98 ± 0.5	32.69 ± 1.65	31.38 ± 1.44	15.58 ± 2.55 ^a^***	22.54 ± 2.12 ^a^***^,b^***	29.89 ± 2.07 ^b^***	29.08 ± 2.67 ^b^***
GSH (µg/g org)	356.47 ± 5.12	368.06 ± 3.98	349.06 ± 6.08	352.05 ± 4.94	194.56 ± 3.34 ^a^***	233.54 ± 5.33 ^a^***^,b^***	370.51 ± 5.45 ^b^***	359 ± 5.78 ^b^***
GPx (nmol GSH/min/mg pr)	11.29 ± 1.59	10.95 ± 0.93	11.56 ± 1.79	10.39 ± 1.23	5.67 ± 1.45 ^a^***	8.67 ± 2.50 ^a^**^, b^**	11.96 ± 0.54 ^b^***	9.49 ± 1.56 ^b^***
MDA (nmol/g rg)	29.81 ± 0.59	25.88 ± 1.15	26.39 ± 2.05	25.09 ± 1.45	70.29 ± 2.19 ^a^***	45.55 ± 1.91 ^a^***^,b^***	31.06 ± 2.19 ^b^***	27.79 ± 2.39 ^b^***

NC: Normal control. NC + L: Normal control treated by leaf extract. NC + B: Normal control treated by bud extract. NC + LB: Normal control treated by the combination of leaf and bud extracts. DC: Diabetic control, Dc+ L: Diabetic + leaf extract. DC + B: Diabetic + buds extract, DC + LB: Diabetic + the combination of leaf and bud extracts. The values are expressed in mean ± SD value, * *p* < 0.05, ** *p* < 0.01, and *** *p* < 0.001; ^a^ comparison between the control group and all other groups; ^b^ comparison between the DC group and the remaining groups.

**Table 6 foods-12-00759-t006:** Effect of the interventions on the oxidative parameters in the pancreas (mean ± SD).

Variables in the Pancreas	Interventions							
	NC	NC + L	NC + B	NC + LB	DC	DC + L	DC + B	DC + LB
Proteins (mg/g org)	4.26 ± 1.05	4.08 ± 1.12	3.89 ± 0.92	4.28 ± 0.94	3.15 ± 0.19 ^a^***	5.23 ± 1.18 ^a^*^, b^***	6.53 ± 0.19 ^b^***	6.27 ± 2.03 ^b^***
Catalase (µmol H_2_ O_2_/min/mg pr)	29.67 ± 3.13	27.97 ± 2.67	30.28 ± 2.78	30.29 ± 1.89	12.29 ± 1.77 ^a^***	25.08 ± 1.72 ^a^*^,b^***	31.71 ± 2.09 ^b^***	29.12 ± 1.89 ^b^***
GSH (µg/g org)	295.93 ± 4.27	289.18 ± 3.78	300.28 ± 5.03	302.05 ± 2.95	100.28 ± 4.08 ^a^***	196.15 ± 4.66 ^a^**^,b^***	312.28 ± 3.88 ^b^***	301.17 ± 2.19 ^b^***
GPx (nmol GSH/min/mg pr)	12.62 ± 1.67	11.92 ± 1.77	12.92 ± 0.69	12.32 ± 1.27	6.19 ± 0.96 ^a^***	10.07 ± 1.89 ^a^*^,b^***	13.09 ± 2.13 ^b^***	11.95 ± 2.02 ^b^***
MDA (nmol/g org)	22.92 ± 3.05	20.34 ± 2.54	20.69 ± 1.92	19.84 ± 2.71	59.29 ± 3.97 ^a^***	31.96 ± 3.18 ^a^***^,b^***	20.15 ± 1.39 ^b^***	21.62 ± 1.37 ^b^***

NC: Normal control. NC + L: Normal control treated by leaf extract. NC + B: Normal control treated by bud extract. NC + LB: normal control treated by the combination of leaf and bud extracts. DC: Diabetic control, DC+ L: Diabetic + leaf extract. DC + B: Diabetic + buds extract, DC + LB: Diabetic + the combination of leaf and bud extracts. The values are expressed in mean ± SD value, * *p* < 0.05, ** *p* < 0.01, and *** *p* < 0.001; ^a^ comparison between the control group and all other groups; ^b^ comparison between the DC group and the remaining groups.

## Data Availability

Data are available upon request.

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
