# Peer review of "Ficus carica (Linn.) Leaf and Bud Extracts and Their Combination Attenuates Type-1 Diabetes and Its Complications via the Inhibition of Oxidative Stress"

_foods, 2023, doi:10.3390/foods12040759_

Round 1

Reviewer 1 Report

The manuscript was designed to evaluate the antioxidant activity and antidiabetic effect of Ficus carica L extracts. Overall, the experimental design is appropriate. The results and discussion are well presented. I think this manuscript needs minor revision. 

1. Line 40,  live with diabetes in 201.  is that correct?

2. Line 202, *p<0.005, is that correct?

3.  In this manuscript, the treatment dose was 200 mg/kg BW. According to the conversion of animal doses to human equivalent doses (HED) based on the body surface area (BSA), what is the result of the daily dose extrapolated to humans?

Author Response

Dear Peer-Reviewers,

We are very thankful for the pertinent comments; we have carefully read the comments and have revised the manuscript accordingly. Our responses are given in a point-by-point manner below. All the changes to the manuscript are highlighted in yellow.

We hope that, in this new form, the manuscript will be suitable for publication in Foods

Reviewer 1:

The manuscript was designed to evaluate the antioxidant activity and antidiabetic effect of Ficus carica L extracts. Overall, the experimental design is appropriate. The results and discussion are well presented. I think this manuscript needs minor revision. 

  1. Line 40, live with diabetes in 201.  is that correct?

Answer: Thank you for your remark, the modification is done, please see line 41 page 1.

  1. Line 202, *p<0.005, is that correct?

Answer: Thank you for your remark, it was a typo, we rectified it, please see line 236 page 5.

  1. In this manuscript, the treatment dose was 200 mg/kg BW. According to the conversion of animal doses to human equivalent doses (HED) based on the body surface area (BSA), what is the result of the daily dose extrapolated to humans?

Answer: Dear reviewer, thank you for your valuable comment, in this study we limited our investigation in animal model (rats).

Applying the following equation:

HED (mg/kg) = Animal does (mg/kg) * Km ratio

In which: Km ratio = Animal Km / human Km

Km (rats)= 6.08 (calculated using the body weight average of our rats 225.5g), and Km (Human)= 37

So HED (mg/kg) =2OO*(6.09/37)

HED (mg/kg) = 32.94 (mg/kg) in humans.

Taking into account the body surface area, and to convert mg/kg to mg/m2 we multiply by human Km.

As a result, the HED (mg/m2) will be 32.94*37 = 1218.91 mg/m2

Please see the reference bellow: “A simple practice guide for dose conversion between animals and human” DOI: 10.4103/0976-0105.177703.

Reviewer 2 Report

ln 40: the date is wrong

References 50. and 59. are repeated

The study present good results with methanolic Ficus carica extracts. Although (under standard laboratory conditions) the toxic effects which are seen in man after exposure to methanol (such as metabolic acidosis and degeneration of the optic nerve) are not observed in experimental animals such as rats because of their different metabolism, this experimental design is not appropriate to treat human diabetes.

It is advised to prepare aqueous or ethanolic extracts and use animal models similar to humans.

Author Response

Dear Peer-Reviewers,

We are very thankful for the pertinent comments; we have carefully read the comments and have revised the manuscript accordingly. Our responses are given in a point-by-point manner below. All the changes to the manuscript are highlighted in yellow.

We hope that, in this new form, the manuscript will be suitable for publication in Foods

Reviewer 2

line 40: the date is wrong

Answer: Thank you for your remark, we rectified it, please see line 41 page 1

References 50. and 59. are repeated

Answer: Thank you for your remark, the modification is done.

The study present good results with methanolic Ficus carica extracts. Although (under standard laboratory conditions) the toxic effects which are seen in man after exposure to methanol (such as metabolic acidosis and degeneration of the optic nerve) are not observed in experimental animals such as rats because of their different metabolism, this experimental design is not appropriate to treat human diabetes.

It is advised to prepare aqueous or ethanolic extracts and use animal models similar to humans.

Answer: Thank you for your valuable remark, in fact, the methanol solvent was used to extract the bioactive compounds soluble in methanol from Ficus carica, then the solvent was evaporated using rotary evaporator to completely eliminate the solvent and to avoid any toxic effect.

Reviewer 3 Report

This work discusses and compares the application of Ficus extract obtained from buds and leaves for amelioration of type-1 diabetes. The manuscript reads well and sufficient
discussion is provided on the obtained results. However, the manuscript should be improved in the Methodology section by inclusion of additional information on the followed protocols.

I have the following comments for improvement of the manuscript:

1. In lines 37-38, a reference should be provided for the statement and is currently missing.

2. In line 40, "201" perhaps the authors wanted to refer to a specific year. Please check and correct.

3. In the reference: https://diabetesatlas.org/

It is stated 537 million in the year 2021, predicted to increase to 643 million in 2030 and 783 million by 2045.

Please check if the stated numbers in the manuscript are presented accurately according to the reference.

4. Keywords should all start with a capital letter.

The punctuation "." should be removed after the keywords.

5. In section 2.6, Provide the name of supplier/breeder of Wistar rats and the age of specified rats.

6. Referring to the explained methodology in section 2.8, prior to explaining the measurement of Catalase activity, in the Methods section, a section should be added explaining the tissue preparation including pancreas, kidney, and liver.

The authors should include the anaesthetic protocol used including the used anaesthetic, its supplier, dosage, supplementation route etc. and explain if the procedure was terminal and rats were killed and how blood samples were collected and tissues were prepared.

7. In lines 211-212 of the Results section, The values are presented in Table 1 in a reverse order different from what is currently explained in the text, 0.26 for buds and 0.31 for leaves. Please check and correct.

Author Response

Dear Peer-Reviewers,

We are very thankful for the pertinent comments; we have carefully read the comments and have revised the manuscript accordingly. Our responses are given in a point-by-point manner below. All the changes to the manuscript are highlighted in yellow.

We hope that, in this new form, the manuscript will be suitable for publication in Foods

Reviewer 3

This work discusses and compares the application of Ficus extract obtained from buds and leaves for amelioration of type-1 diabetes. The manuscript reads well and sufficient
discussion is provided on the obtained results. However, the manuscript should be improved in the Methodology section by inclusion of additional information on the followed protocols.
I have the following comments for improvement of the manuscript:
1. In lines 37-38, a reference should be provided for the statement and is currently missing.

Answer: thank you for your comment, the reference required was added, please see line 39 and 40 page1.  
2. In line 40, "201" perhaps the authors wanted to refer to a specific year. Please check and correct.

Answer: Thank you for your remark, the modification is done, please see line 41 page1. 

3. In the reference: https://diabetesatlas.org/It is stated 537 million in the year 2021, predicted to increase to 643 million in 2030 and 783 million by 2045.
Please check if the stated numbers in the manuscript are presented accurately according to the reference.

Answer: the numbers are rechecked and verified; the numbers documented in the first version are correct. A reference which treated the diabetes prevalence estimates for 2021 and projections for 2045 was missing and we have added it.

  1. Keywords should all start with a capital letter. The punctuation "." should be removed after the keywords.

Answer: Thank you for your remark, the modification is done.

  1. In section 2.6, Provide the name of supplier/breeder of Wistar rats and the age of specified rats.

Answer: Thank you for your remark, the modification is done, please see from 155 to 163 page 4.
6. Referring to the explained methodology in section 2.8, prior to explaining the measurement of Catalase activity, in the Methods section, a section should be added explaining the tissue preparation including pancreas, kidney, and liver.

The authors should include the anaesthetic protocol used including the used anaesthetic, its supplier, dosage, supplementation route etc. and explain if the procedure was terminal and rats were killed and how blood samples were collected and tissues were prepared.

Answer: Thank you for your remark, the modification is done. We have added a section before the ‘Biochemical Analysis’ entitled: blood sampling and organ preparation, please see from line 194 to line 205-page 4and 5.

Reference: https://www.ncbi.nlm.nih.gov/pmc/articles/PMC3989930/

The protocol used to scarify rats, and for the collection of tissues was added in the manuscript, according to the bellow reference, please see from line 206 to line 211 page 5.

Reference: https://linkinghub.elsevier.com/retrieve/pii/S0188440918303242

  1. In lines 211-212 of the Results section, the values are presented in Table 1 in a reverse order different from what is currently explained in the text, 0.26 for buds and 0.31 for leaves. Please check and correct.

Answer: Thank you for your valuable remark, the modification id done. Please see 245-246 page 5.

Round 2

Reviewer 2 Report

All corrections and explanations made